# Early Introduction of Allergenic Foods and the Prevention of Food Allergy

**DOI:** 10.3390/nu14132565

**Published:** 2022-06-21

**Authors:** Brit Trogen, Samantha Jacobs, Anna Nowak-Wegrzyn

**Affiliations:** 1Department of Pediatrics, NYU Grossman School of Medicine, Hassenfeld Children’s Hospital, New York, NY 10016, USA; samantha.jacobs@nyulangone.org; 2Allergy and Immunology, Department of Pediatrics, NYU Grossman School of Medicine, Hassenfeld Children’s Hospital, New York, NY 10016, USA; anna.nowak-wegrzyn@nyulangone.org; 3Department of Pediatrics, Gastroenterology and Nutrition, Collegium Medicum, University of Warmia and Mazury, 10-719 Olsztyn, Poland

**Keywords:** allergy, food allergy, atopic dermatitis, eczema, atopy, prevention, sensitization, oral tolerance, early food introduction

## Abstract

The increasing prevalence of food allergies is a growing public health problem. For children considered high risk of developing food allergy (particularly due to the presence of other food allergies or severe eczema), the evidence for the early introduction of allergenic foods, and in particular peanut and egg, is robust. In such cases, the consensus is clear that not only should such foods not be delayed, but that they should be introduced at approximately 4 to 6 months of age in order to minimize the risk of food allergy development. The early introduction of allergenic foods appears to be an effective strategy for minimizing the public health burden of food allergy, though further studies on the generalizability of this approach in low-risk populations is needed.

## 1. Introduction

Although clear epidemiologic data are lacking, food allergies are believed to affect up to 10% of individuals in Westernized countries [1]. The prevalence of food allergy overall, and of peanut allergy in particular, appears to be steadily increasing, including in developing countries [1,2]. Food allergies can result in significant morbidity and psychosocial burden (including the risk of nutritional deficiencies and life-threatening anaphylaxis), as well as high costs for the healthcare system and for families of food-allergic children [3,4]. As a result, food allergies can have a significant negative impact on quality of life [5]. Given the significant morbidity associated with these conditions and their inability to be cured, preventing the development of food allergies before they begin is essential.

Substantial research has aimed to identify primary prevention strategies for food allergy. Many interventions that have been attempted in pregnant or breastfeeding women and infants appear to have little to no benefit in preventing food allergy, including dietary avoidance of food allergens, vitamin supplements, fish oil, probiotics, prebiotics, and synbiotics—however, it should be noted that the evidence remains uncertain in many cases [6]. Optimal skin care and aggressive early treatment of atopic dermatitis using emollients, while thought to enhance skin barrier function, has also failed to show significant impact in the later development of food allergy, though some have argued that studies have not targeted high-risk infants or the use of optimal emollient methods [6,7,8,9]. In this setting, early introduction of allergenic foods in infancy has emerged as one of the more promising strategies to decrease food allergy development.

## 2. Evolution of the Guidelines for Food Allergy Prevention

Until 2008, clinical practice guidelines from the American Academy of Pediatrics (AAP) and other professional societies recommended delaying the introduction of allergenic foods, such as peanut, until 3 years of age. This recommendation was based on the theory that the lack of exposure to allergenic foods during early infancy—which was posited to be a developmental window of high susceptibility—would prevent later development of allergy. Though well-intentioned, it is now clear that these guidelines may have contributed to, rather than prevented, the development of food allergies in children. The dual exposure hypothesis offers one possible explanation as to why this approach failed, proposing that allergen exposure through skin may lead to IgE sensitization unless oral tolerance is first induced through the gastrointestinal tract [10]. Children with eczema are known to be at higher risk of developing peanut and egg allergy, possibly due to the increased likelihood of sensitization, due to skin barrier disruption, see Figure 1 [1]. Both clinical observation and animal studies also support the “outside-in” model of epicutaneous food allergen sensitization, suggesting that later introduction of allergens in the diet may occur too late, after allergens have already been introduced via the skin or respiratory tract [9]. In response to these findings, many food allergy prevention guidelines now recommend the early introduction of allergenic foods, such as peanut and egg, as part of complementary feeding in infancy, see Table 1 [2].

**Figure 1 nutrients-14-02565-f001:**
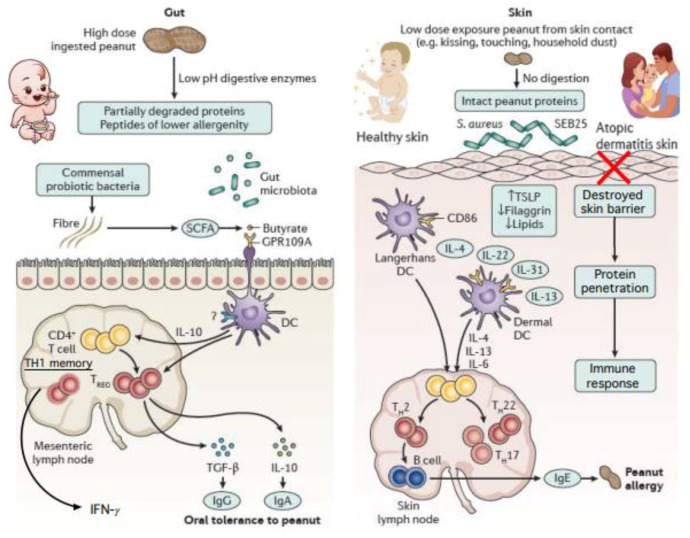
Exposure to foods via ingestion results in oral tolerance in contrast to cutaneous or inhalational exposures that promote IgE-sensitization [11].

**Table 1 nutrients-14-02565-t001:** A summary of international consensus guidelines on early introduction of allergenic foods.

Professional Body	Publication Year	Recommendations
American Academy of Pediatrics (AAP) [12]	2019	High-risk infants (presence of severe eczema and/or egg allergy) should be introduced to peanut as early as 4–6 months of age, following successful feeding of other solid food(s) to ensure the infant is developmentally ready. Allergy testing is strongly advised prior to peanut introduction for this group. Infants with mild-to-moderate eczema should be introduced to peanut around 6 months of age, in accordance with family preferences and cultural practices, to reduce the risk of peanut allergy Infants without eczema or food allergy who are not at increased risk: peanut should be introduced freely into the diet together with other solid foods and in accordance with family preferences and cultural practices.
Asia Pacific Association of Pediatric Allergy, Respirology & Immunology (APAPARI) [13]	2017	Healthy infants: Introduce complementary foods at 6 months of age. At-risk infants (family history of atopy): No delay in introduction of allergenic foods. To be introduced in a sensible manner once weaning has commenced. High-risk infants with severe eczema: Allergy testing to egg (and peanut in countries with high peanut allergy prevalence). Supervised oral challenges in sensitized infants, followed by introduction of the allergenic food into the infant’s regular diet if challenge negative. Introduction of all allergenic foods should not be delayed. Aggressive control of eczema.
Australian Society of Clinical Immunology and Allergy (ASCIA) [14]	2020	At around six months, but not before four months, start to introduce a variety of solid foods, starting with iron rich foods, while continuing breastfeeding. All infants should be given allergenic solid foods including peanut butter, cooked egg, dairy, and wheat products in the first year of life. This includes infants at high risk of allergy.
Canadian Paediatric Society (CPS) [15]	2021	For high-risk infants, encourage the introduction of allergenic foods (e.g., cooked (not raw) egg, peanut) early, at about 6 months and not before 4 months of age, in a safe and developmentally appropriate way, at home. In infants at low risk for food allergy, allergenic foods can also be introduced at around 6 months of age. When allergenic foods have been introduced, make sure that ongoing ingestion of age-appropriate serving sizes is regular (i.e., a few times a week), to maintain tolerance.
European Academy of Allergy and Immunology (EAACI) [16]	2020	The EAACI Task Force suggests introducing well-cooked hen’s egg, but not raw egg or uncooked pasteurized egg, into the infant diet as part of complementary feeding to prevent egg allergy in infants. In populations where there is a high prevalence of peanut allergy, the EAACI Task Force suggests introducing peanuts into the infant diet in an age-appropriate form as part of complementary feeding in order to prevent peanut allergy in infants and young children. The EAACI Task Force suggests avoiding supplementing with cow’s milk formula in breastfed infants in the first week of life to prevent cow’s milk allergy in infants and young children (low quality evidence).
German Society for Allergology and Clinical Immunology (DGAKI) [17]	2014	The current recommendation in Germany to introduce solid foods to infants over the age of 4 months is reasonable given increasing nutritional requirements. The introduction of solid foods should not be delayed as a means of allergy prevention. There is no evidence to suggest that dietary restriction in the form of avoiding potent food allergens in the first year of life has a preventive effect. Such a measure is therefore not recommended. There is currently no reliable evidence that the introduction of potent food allergens during the first 4 months of life has a preventive effect. There is evidence that a child’s consumption of fish during the first year of life has a protective effect against the development of atopic diseases. Fish should be introduced in solid foods.
National Institute of Allergy and Infectious Diseases (NIAID) [18]	2017	For infants with severe eczema, egg allergy, or both: Strongly consider evaluation by sIgE measurement and/or SPT and, if necessary, an OFC. Based on test results, introduce peanut-containing foods as early as 4 to 6 months of age to reduce the risk of peanut allergy. For infants with mild-to-moderate eczema: introduce age-appropriate peanut-containing food around 6 months of age. For infants without eczema or any food allergy: introduce age-appropriate peanut-containing foods freely together with other solid foods and in accordance with family preferences and cultural practices.

### 2.1. Peanut

The landmark Learning Early About Peanut (LEAP) study published in 2015 was the first to suggest that the early introduction of allergenic foods decreases the incidence of food allergy, leading to a paradigm shift away from early food avoidance [19]. LEAP demonstrated that the introduction of peanut in high-risk infants between the ages of 4–6 months decreased the prevalence of IgE-mediated peanut allergy at 5 years of age by over 80% when compared to introduction after 12 months of age [19]. High-risk infants in this study were defined as those with severe eczema and/or egg allergy.

In a follow-up study entitled the Persistence of Oral Tolerance to Peanut (LEAP-On), children who consumed peanuts from infancy through to age five followed by one year of peanut avoidance were 74% less likely to have peanut allergy than children who had consistently avoided peanuts up until age six, suggesting that the tolerance induced by early introduction can persist even in the absence of repeated exposures. Later analysis of the LEAP study cohort showed that early introduction of peanut did not negatively impact growth, nutrition, or duration of breastfeeding [20]. In addition, early introduction of peanut was found to be allergen-specific, and had no impact on the development or resolution of other allergic diseases, including asthma and atopic dermatitis [21]. Subsequent studies also identified additional independent risk factors for the development of peanut allergy in the context of peanut avoidance, including genetic susceptibility (via the MALT1 gene and HLA alleles) and *Staphylococcus aureus* colonization [22,23,24]. These findings are suggestive of the multiple environmental, genetic, epigenetic, and social factors at play in the development of food allergy.

### 2.2. Egg

Trials examining the early introduction of eggs as a means of reducing egg allergy have yielded mixed results. In 2017, the Prevention of Egg Allergy with Tiny Amount Intake Trial (PETIT) randomized 147 infants with atopic dermatitis to consume either heated egg powder or placebo, in addition to undergoing the aggressive treatment of atopic dermatitis [25]. When compared to the avoidance of egg for the first year of life, those infants who consumed egg powder from 6 to 12 months of age had a significant reduction in the development of egg allergy (8% of the egg group, compared with 38% of the placebo group), resulting in the trial being stopped early to avoid harm to the placebo group [25].

In contrast to these positive findings, the Solids Timing for Allergy Research (STAR) trial randomized high-risk infants with moderate-to-severe eczema to receive either whole egg powder or rice powder daily from 4 to 8 months of age, followed by cooked egg from 8 months onward [26]. Although there was a non-significant trend towards reduced egg allergy in the egg ingestion group, this trial was terminated early due to the high rates of allergic reactions in the egg powder group. It was also noted that 36% of infants in this study had high levels of egg-specific IgE at 4 months of age even in the absence of known egg exposure, suggesting that pre-existing sensitization may be common in this high-risk group [26].

In the 2017 Beating Egg Allergy Trial (BEAT), 319 infants with at least one first-degree relative with an allergic disease were randomized to receive whole-egg powder or placebo from 4 to 8 months of age, after which diets were liberalized in both groups [27]. At 12 months of age, the absolute risk reduction the development of a positive egg white skin prick test was 9.8%, though there was no statistically significant decrease in the proportion of children with a probable egg allergy.

In the Hen’s Egg Allergy Prevention (HEAP) study, 383 infants aged 4 to 6 months with confirmed negative skin prick tests prior to trial initiation were randomized to receive either egg white powder (equivalent to raw egg whites) or placebo three times a week until 12 months of age [28]. No statistically significant reduction in rates of egg allergy was observed in this trial. Rather, this study was concerning due to the fact that of infants initially excluded from the trial due to positive skin prick testing, two-thirds developed anaphylaxis when subsequently introduced to egg via oral challenge to raw egg white powder. Given that such test results are rarely available in real-world settings, these findings raise concern for the universal implementation of early egg introduction using raw egg white powder.

### 2.3. Milk

Several studies suggest that early exposure to cow’s milk may prevent later development of cow’s milk protein allergy. In 2010, an observational study of over 13,000 Israeli infants from a population-based birth cohort found that early and regular exposure to cow’s milk-based formula (beginning in the first two weeks of life) was associated with significantly decreased rates of milk allergy by age 3–5 years in comparison to those introduced to formula after 3 months of age [29].

In the HealthNuts Study, an observational longitudinal study of 5276 infants, early exposure to cow’s milk in the first 3 months of life was associated with the decreased risk of cow’s milk sensitization (measured via skin prick testing) and parent-reported cow’s milk allergy at one year of age [30]. No significant differences were noted in risk of other food allergies.

The timing of milk introduction appears especially important given the widespread use of supplementary cow’s milk-based formula in the first days and weeks of life. In a 2016 case-control study of 185 infants, the early introduction of cow’s milk formula was associated with a lower incidence of IgE-mediated cow’s milk allergy in comparison to infants who received delayed (after one month of age) or no formula [31]. However, in the Atopy Induced by Breastfeeding or Cow’s Milk Formula (ABC) study, temporary cow’s milk formula supplementation in the first 3 days of life with subsequent removal from the infant diet was associated with an increased risk of milk allergy and anaphylaxis in early childhood [6]. When consumed on a regular basis, cow’s milk formula did not appear to increase the risk of milk allergy in this study [6].

### 2.4. Multiple Foods

Some trials have examined the early introduction of multiple allergenic foods simultaneously. In the 2016 Enquiring About Tolerance (EAT) Study, 1303 infants were randomized to the introduction of six allergenic foods (peanut, cooked egg, cow’s milk, sesame, whitefish, and wheat) at either 3 months (early introduction) or 6 months of age (standard) [32]. Although the study was limited by high rates of non-adherence to dietary protocols, a significant reduction in the prevalence of peanut and egg allergy was observed in the subset of the early introduction group that consumed at least 2 g of each food protein per week [32]. In the primary analysis, a 20% reduction in food allergy overall was also observed in the early introduction group, though these results were not statistically significant [32]. An additional analysis found that introduction of gluten between the ages 4 and 6 months was associated with reduced prevalence of celiac disease [33]. A secondary intention-to-treat analysis of this trial conducted in 2019 found that early introduction was effective in preventing food allergies among a subset of infants considered high risk for allergy development: those with food sensitization at enrollment, and those with visible eczema at enrollment [34]. Of note, the early introduction of solids was also not associated with the decreased rates of breastfeeding in this study, but did result in small but significant improvements in infant sleep duration and quality [35,36].

In a similar vein, the SEED trial was a randomized controlled trial in which 163 infants aged 3–4 months with atopic dermatitis were randomized to receive either mixed allergenic food powder (containing egg, milk, wheat, soybean, buckwheat, and peanuts) or placebo powder [37]. The amount of powder was gradually increased over the trial and continued for 12 weeks. Following this intervention, a significant difference was noted in the incidence of food allergy episodes (RR 0.301, 95% CI 0.116–0.784, *p* = 0.0066) and egg allergies at 18 months of age [37]. The Tolerance Induction Through Early Feeding to Prevent Food Allergy in Infants with Eczema (TEFFA) study is a randomized controlled trial that will build on these findings by examining whether the early introduction of hen’s egg, cow’s milk, peanut, and hazelnut can reduce the risk of developing food allergies in the first year of life in children with atopic dermatitis [38].

## 3. Study Generalizability

A question remains regarding the efficacy of the early introduction of allergenic foods for preventing food allergy in a general population with no underlying risk factors for allergy. Some have argued that the evidence supporting the efficacy of early food introduction in the general (i.e., not high-risk) population is less compelling given that the intention-to-treat analysis of both HEAP and EAT failed to show a statistically significant decrease in food allergies in a general population [39]. Although per-protocol analysis in EAT did show a statistically significant reduction in food allergy, the potential for bias to be influencing these results cannot be discounted. In a 2020 systematic review, the lack of uniformity in study methodology and patient population was cited as a limitation in both the generalizability of study conclusions as well as accurate subgroup analysis [6].

## 4. Barriers to Adherence

Multiple barriers may prevent the successful implementation of food allergy prevention programs based on early food introduction [40]. Issues with respect to patient education, access to health services, child cooperation, parental fears or perceived low self-efficacy, and practical difficulties adhering to complicated or long-term treatment plans have been identified as interfering with early food introduction programs [41]. In the EAT study, nonwhite ethnicity, increased maternal age, and infants with feeding difficulties and early onset eczema were all associated with decreased protocol adherence [32,42].

Addressing health disparities based on race and ethnicity is especially important given that Black children are at greater risk of developing food allergies in comparison to other racial groups, and that allergies are increasing in prevalence in this population [43]. Black food-allergic patients are also more likely to experience heightened psychosocial burdens and negative health outcomes related to food allergies [43]. Understanding and addressing these racial and ethnic disparities must be prioritized in future guidelines and public health programs aimed at preventing food allergy. Like food allergen exposure itself, patient and community involvement should occur early and be sustained throughout the development of such programs if they are to be effective [40].

## 5. Other Directions in Food Allergy Prevention

A number of other strategies are under active investigation for the prevention of food allergy. At this time, there are 14 active clinical trials listed on clinicaltrials.gov investigating probiotics, diet enriched in *Prevotella* spp. and butyrate, vaginal seeding, Mediterranean diet in pregnancy, vitamin D supplementation, and intensive restoration of skin barrier function in atopic dermatitis [44]. An overview of the potential alternative targets for food allergy prevention is also illustrated in Figure 2.

**Figure 2 nutrients-14-02565-f002:**
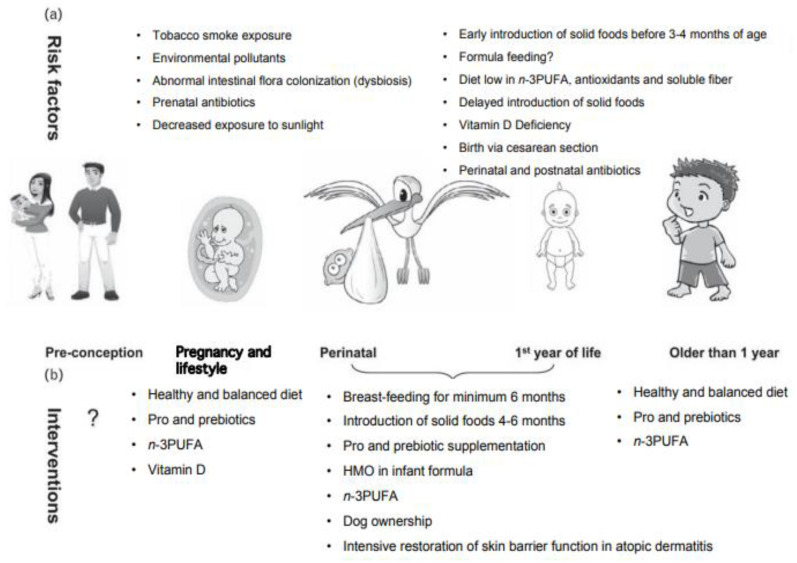
Other directions currently being investigated for food allergy prevention; (**a**) risk factors; (**b**) interventions. Modified with permission from Prescott and Nowak-Węgrzyn [45]. Legend: n-3PUFA: n-3 Polyunsaturated Fatty Acids, which can include eicosapentaenoic acid (EPA) and docosahexaenoic acid (DHA) can provide multiple health benefits including anti-inflammatory effects [46]. HMO: Human milk oligosaccharides, a naturally present constituent of human milk that can modulate the immune system [47].

## 6. Conclusions

The increasing prevalence of food allergies around the world is a growing global public health problem. Recent research suggests that while cutaneous or inhalational exposures of allergens may promote IgE-sensitization, ingestion of allergenic foods can result in oral tolerance [46]. As a result, many health organizations have updated their guidelines away from allergen avoidance in early infancy in favor of early exposure. The early introduction of allergenic foods appears to be an effective strategy for minimizing the population burden of food allergy, though further studies on the generalizability of this approach in low-risk populations are needed. For children considered at high risk of developing food allergy (particularly due to the presence of other food allergies or severe eczema), the evidence for the early introduction of allergenic foods, and in particular peanut and egg, is robust. In such cases, the consensus is clear that not only should such foods not be delayed, but that they should be introduced at approximately 4 to 6 months of age in order to minimize the risk of food allergy development. The many clinical trials currently underway examining food allergy prevention will further strengthen our understanding of the complex process of food allergy development.

In line with the dual exposure hypothesis, high-dose exposure to ingested food protein promotes development of oral tolerance, whereas low-dose exposure through inflamed skin (disrupted epithelial barrier) or via inhalation is more likely to prime for IgE-sensitization to food proteins. Modified from Nowak-Wegrzyn, Szajewska, and Lack [11].

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
