# Peer review of "Early Introduction of Allergenic Foods and the Prevention of Food Allergy"

_nutrients, 2022, doi:10.3390/nu14132565_

Round 1
Reviewer 1 Report
I reviewed the manuscript ‘Early Introduction of Allergenic Foods and the Prevention of Food Allergy’ by Brit Trogen, et al as you requested. The paper was thorough, well written, and well organized. The table provided a though summary of recommendations by important professional bodies concerning the early introduction of allergenic foods into the diet. In addition, the two figures were well designed and applicable to the presentation. The 62 references were current, pertinent, and carefully discussed. I did not detect any errors in the manuscript. The authors also provided thoughtful comments regarding future research on the prevention of food allergies. In particular, it is still uncertain concerning the prevention of food allergies in low risk populations. Finally, the authors discussed the barriers to the implementation to early Introduction of allergenic foods in the prevention of food allergies.
Author Response
Thank you for your kind feedback
Reviewer 2 Report
The paper is interesting to the field, and clearly written.
However, I suggest to include some more papers to have it more comprehensive. Otherwise, I have only some minor remarks:
* Abstract: would put line 15-16 as concluding sentence of the abstract
* Introduction: please cite Brough H (DOI: 10.1111/all.15006) for other potentially important influence factors; I think moisturizers are discussed differently there
* line 51: discussin more detail why dual exposure hypothesis explains that later food introduction is counter-productive (e.g. "do children without allergens in their early diet meet the proteins really via skin and respiratory tract and therefore via the wrong route?)
* line 58 58/59: sentence can be deleted, just reference to Table 1
* Table 1: German guidelines "S3 Allergy Prevention" could also be included (DOI: 10.1111/all.15006)
* Table 1: please make clear sections (put in lines) between different papers
* Paper by Fleischer et al could also be included (https://doi.org/10.1111/pde.12685)
* Table 1: EAACI paper: certainty of evidence for avoiding cow's milk formula in first week of life is classified as "low"; please add this info to table
* line 80: early introduction of peanut was found to be allergen-specific
* line 92: "147 infants" with atopic dermatitis?
* line 96: "trial being stopped early" - why?
* line 99: "in contrast" to what?
* line 123: "raise concern"..but are the results not only relevant for SPT-positive children rather than "universal implementation"?
* line 131: "milk allergy" at which age?
* Figure 1: scheme of children on right panel are unclear, please use "kissing" or "touching" or "cream" for visualization; left panel bottom: "IFN-" typo? right panel below skin: would rather use "destroyed skin barrier => protein penetration => immune response"
* Figure 2: put explantation of abbrev. into figure legends
Author Response
* Abstract: would put line 15-16 as concluding sentence of the abstract
- Thank you, we revised abstract as suggested.
* Introduction: please cite Brough H (DOI: 10.1111/all.15006) for other potentially important influence factors; I think moisturizers are discussed differently there
- Added citation with note: “though some have argued that studies have not targeted high-risk infants or the use of optimal emollient methods.”
* line 51: discussion more detail why dual exposure hypothesis explains that later food introduction is counter-productive (e.g. "do children without allergens in their early diet meet the proteins really via skin and respiratory tract and therefore via the wrong route?)
- Added comment: “Both clinical observation and animal studies also support the “outside-in” model of epicutaneous food allergen sensitization, suggesting that later introduction of allergens in the diet may occur too late, after allergens have already been introduced via the skin or respiratory tract.”
* line 58 58/59: sentence can be deleted, just reference to Table 1
- Done
* Table 1: German guidelines "S3 Allergy Prevention" could also be included (DOI: 10.1111/all.15006)
- German guidelines have been added to Table 1.
* Table 1: please make clear sections (put in lines) between different papers
- As this is a stylistic preference we will defer to the journal’s style guide (there are currently shaded lines separating different guidelines), and if the journal has no preference prefer the current style to adding lines.
* Paper by Fleischer et al could also be included (https://doi.org/10.1111/pde.12685)
- This is a consensus report of many of the guidelines already listed in the table, and offers redundant information at it is derived from the guidelines already included.
* Table 1: EAACI paper: certainty of evidence for avoiding cow's milk formula in first week of life is classified as "low"; please add this info to table
- Added the note: “(low quality evidence)”
* line 80: early introduction of peanut was found to be allergen-specific
- Unclear what this comment refers to, as this is stated in the paper.
* line 92: "147 infants" with atopic dermatitis?
- Added a note specifying “147 infants with atopic dermatitis”
* line 96: "trial being stopped early" - why?
- Added a note: “resulting in the trial being stopped early to avoid harm to the placebo group.”
* line 99: "in contrast" to what?
- Added a note: “In contrast to these positive findings”
* line 123: "raise concern"..but are the results not only relevant for SPT-positive children rather than "universal implementation"?
- Yes, this is the point. As noted, given that SPT is not universally implemented prior to egg introduction, implementing early introduction universally would risk a negative outcome among those who would have had a positive SPT but have never actually received one.
* line 131: "milk allergy" at which age?
- Added a note: “milk allergy by age 3-5 years”
* Figure 1: scheme of children on right panel are unclear, please use "kissing" or "touching" or "cream" for visualization; left panel bottom: "IFN-" typo? right panel below skin: would rather use "destroyed skin barrier => protein penetration => immune response"
- Revised, as suggested
* Figure 2: put explanation of abbrev. into figure legends
- Revised as suggested
